# Label-Free Quantitative Acetylome Analysis Reveals *Toxoplasma gondii* Genotype-Specific Acetylomic Signatures

**DOI:** 10.3390/microorganisms7110510

**Published:** 2019-10-30

**Authors:** Ze-Xiang Wang, Rui-Si Hu, Chun-Xue Zhou, Jun-Jun He, Hany M. Elsheikha, Xing-Quan Zhu

**Affiliations:** 1State Key Laboratory of Veterinary Etiological Biology, Key Laboratory of Veterinary Parasitology of Gansu Province, Lanzhou Veterinary Research Institute, Chinese Academy of Agricultural Sciences, Lanzhou 730046, China; neau601@126.com (Z.-X.W.); grishu0707@gmail.com (R.-S.H.); zhouchunxue23@163.com (C.-X.Z.); hejunjun@caas.cn (J.-J.H.); 2College of Veterinary Medicine, Gansu Agricultural University, Lanzhou 730070, China; 3College of Animal Science and Technology, Jilin Agricultural University, Changchun 130118, China; 4Department of Parasitology, Shandong University School of Basic Medicine, Jinan 250012, China; 5Faculty of Medicine and Health Sciences, School of Veterinary Medicine and Science, University of Nottingham, Sutton Bonington Campus, Loughborough LE12 5RD, UK

**Keywords:** *Toxoplasma gondii*, genotype, tachyzoite, acetylation, label-free

## Abstract

Distinct genotypic and pathogenic differences exist between *Toxoplasma gondii* genotypes. For example, genotype I is highly virulent, whereas genotype II and genotype III are less virulent. Moreover, Chinese 1 genotype (ToxoDB#9) is also virulent. Here, we compare the acetylomes of genotype 1 (RH strain) and Chinese 1 genotype (ToxoDB#9, PYS strain) of *T. gondii*. Using mass spectrometry enriched for acetylated peptides, we found a relationship between the levels of protein acetylation and parasite genotype-specific virulence. Notably, lysine acetylation was the largest (458 acetylated proteins) in RH strain, followed by PYS strain (188 acetylated proteins), whereas only 115 acetylated proteins were detected in PRU strain. Our analysis revealed four, three, and four motifs in RH strain, PRU strain and PYS strain, respectively. Three conserved sequences around acetylation sites, namely, xxxxxK^Ac^Hxxxx, xxxxxK^Ac^Fxxxx, and xxxxGK^Ac^Sxxxx, were detected in the acetylome of the three strains. However, xxxxxK^Ac^Nxxxx (asparagine) was found in RH and PYS strains but was absent in PRU strain. Our analysis also identified 15, 3, and 26 differentially expressed acetylated proteins in RH strain vs. PRU strain, PRU strain vs. PYS strain and PYS strain vs. RH strain, respectively. KEGG pathway analysis showed that a large proportion of the acetylated proteins are involved in metabolic processes. Pathways for the biosynthesis of secondary metabolites, biosynthesis of antibiotics and microbial metabolism in diverse environments were featured in the top five enriched pathways in all three strains. However, acetylated proteins from the virulent strains (RH and PYS) were more enriched in the pyruvate metabolism pathway compared to acetylated proteins from PRU strain. Increased levels of histone-acetyl-transferase and glycyl-tRNA synthase were detected in RH strain compared to PRU strain and PYS strain. Both enzymes play roles in stress tolerance and proliferation, key features in the parasite virulence. These findings reveal novel insight into the acetylomic profiles of major *T. gondii* genotypes and provide a new important resource for further investigations of the roles of the acetylated parasite proteins in the modulation of the host cell response to the infection of *T. gondii*.

## 1. Introduction

*Toxoplasma gondii* is an apicomplexan protozoan parasite which has caused significant morbidity and mortality worldwide [1]. This parasite can infect nearly all vertebrate species, including mammals, birds, and humans. In immune-competent individuals, infection with *T. gondii* is usually asymptomatic or causes latent infection. However, the latent infection can be reactivated in immunocompromised individuals, leading to adverse health consequences such as encephalitis [2]. *T. gondii* exhibits a complex life-cycle, which includes the asexual reproductive phase occurring in the intermediate host and sexual phase that occurs in the intestine of the definitive feline host. The asexual reproduction results in the formation of fast-replicating tachyzoites, which play a key role in the dissemination of *T. gondii* infection during acute infection. Tachyzoites can be differentiated into slow-replicating bradyzoites enclosed within cystic structures, in response to stress. The sexual reproduction in the feline intestine ends up with the formation and shedding of environmentally resistant oocysts.

*T. gondii* encompasses four archetypal lineages (type I, II, III, and 12) in North America and Europe. These clonal lineages exhibit remarkable differences in terms of virulence, migration and growth rate [2]. Type I strains are highly virulent (100% lethal dose (LD_100_), 1 parasite), whereas types II, III, and 12 are less virulent (LD_50s_ of ~10^3^, ~10^5^, and ~10^3^ parasites, respectively) [3,4]. The virulent PYS belongs to the main clonal lineage in China Chinese 1 (ToxoDB#9) [5,6]. High-throughput ‘omics’ methodologies have been powerful tools to characterize the genomes, transcriptomes, and proteomes of *T. gondii* strains belonging to different genotypes [7,8,9,10] and have played an essential role in the progression of *T. gondii* research in the postgenomic era [10,11,12,13,14,15]. Advances in mass spectrometry (MS)-based proteomics and high-affinity purification of acetylated peptides have enabled the characterization of lysine acetylation [16].

Lysine acetylation is a reversible protein post-translational modification (PTM), which plays key roles in crucial biological processes, such as mRNA stability, enzyme activity, physiochemical properties of proteins, metabolism regulation, and protein–protein or protein–nucleic acid interactions [16,17,18,19]. Lysine acetylation has been studied in several parasites, such as *Plasmodium falciparum*, *Schistosoma japonicum* and *T. gondii* [20,21,22]. This process has been detected in proteins involved in diverse cellular functions in *T*. *gondii* and modulation of acetylation pathways were suggested as a potential target for anti-parasite drug discovery [23]. However, information about this PTM in *T. gondii* is only available for tachyzoites of type I RH strain and the acetylomes of other *T. gondii* strains are still unknown. Fortunately, the availability of the genome of multiple genotypes of *T. gondii* provides a valuable resource for the identification of lysine-acetylated sites and proteins in the different parasite genotypes.

In the present study, we investigated whether differences exist in lysine acetylation between *T. gondii* strains of different genotypes/virulence abilities. The lysine acetylomes of virulent *T. gondii* strains, namely RH (type I) strain, PYS (Chinese 1), and moderate virulent PRU (type II) strain, were determined using antiacetyl-lysine immunoprecipitation to enrich the acetylpeptides, followed by the LC–MS/MS analysis. Further intergenotype comparison of the reported *T. gondii* acetylomes revealed genotype-specific differences in key virulence mechanisms such as stress tolerance. The study findings provide new resources for further functional analysis of lysine acetylation in *T. gondii*.

## 2. Material and Methods

### 2.1. Ethics Approval and Consent to Participate

All animal experiments have been performed with the permission of the Animal Administration and Ethics Committee of Lanzhou Veterinary Research Institute, Chinese Academy of Agricultural Sciences (Permit No. LVRIAEC-2016-06), and according to the Animal Ethics Procedures and Guidelines of the People’s Republic of China. Appropriate measures were taken to minimize the suffering of the animals during the experiments.

### 2.2. Parasite Strains and Mice

RH (Type I) strain, PRU (Type II) strain and PYS (Chinese 1) strain of *T. gondii* were maintained in our laboratory. Female, six-to-eight-week-old, specific-pathogen-free (SPF) BALB/c mice were obtained from the Laboratory Animal Center of Lanzhou Veterinary Research Institute.

### 2.3. Parasite Preparation

Although some researchers prepared tachyzoites of *T. gondii* using tissue culture systems, studies on tachyzoites prepared using mice inoculation are necessary. The sensitivity of tissue culture and mouse inoculation methods for the demonstration of *T. gondii* organisms were found to be equal [24]. In addition, many studies prepared tachyzoites using mice inoculation [15,25,26,27,28,29]. The purpose of our research was to investigate acetylomic signatures of different genotypes of *Toxoplasma gondii* purified *in vivo*. Tachyzoites of RH and PYS strains were recovered from frozen stocks and passaged in BALB/c mice for three generations in order to allow them to regain their virulence [15,27]. Mice were infected with tachyzoites (~10^2^). Several days later, mice showed symptoms of toxoplasmosis, such as ruffled fur, reduced appetite, and head tilting. Then, mice were humanely euthanized, and their body cavities were washed with sterile phosphate-buffered saline (PBS) to harvest the tachyzoites. The peritoneal fluid containing the tachyzoites was pelleted by centrifugation for 15 min at 1680× *g* and the pellet was washed three times with PBS. The pellet was digested with trypsin (0.25%) at 37 °C for 20 min and centrifuged for 15 min at 1680× *g* to remove murine cells and proteins. Finally, the pellet was suspended in PBS and preserved frozen in an Eppendorf tube at −80 °C until analysis. For PRU strain, it is an avirulent strain and cyst-forming in mice [30]. To isolate the PRU tachyzoites, mice were treated with dexamethasone (0.2 mg per mouse) on alternate days, three times [15,28]. After the third injection, mice were orally inoculated with 100–150 cysts of PRU strain and were continued to receive dexamethasone every other day after the infection. Nine days after infection, mice started to exhibit symptoms of toxoplasmosis. Then, mice were humanely euthanized and peritoneal fluid was harvested, and tachyzoites were enriched by centrifugation as described above. The number of the purified tachyzoites of RH, PRU, and PYS strains was determined using a hemocytometer.

### 2.4. Protein Extraction and Digestion

Total protein was extracted from an equal number of tachyzoites (~5 × 10^8^) from RH, PRU and PYS strains. Two biological replicates were prepared for each parasite strain. Briefly, tachyzoites were transferred to a protein lysis buffer (8 M urea, pH 8.0), containing protease inhibitor (PMSF, 1 mM, Thermo Scientific, Chelmsford, MA, USA) and deacetylase inhibitors (100 Mm Tris pH 8.0, 1 mM para-amino-benzamidine, 5 mM caproic acid, 2 mM leupeptin, 5 µM PTACH, 2 µg/mL apicidin, Sigma-Aldrich Co. LLC, St. Louis, MA, USA), and were lysed by sonication on ice (2/3 s, 5 min) using a high-intensity ultrasonic processor (Scientz Biotechnology Co. LTD, Ningbo, Zhejiang, China). The lysate was centrifuged at 20,000x *g* for 20 min at 4 °C (Eppendorf, Hamburg, Germany) in order to remove debris. After centrifugation, the supernatant was treated with 10 mM dithiothreitol (DTT, Amersco, Solon, State of OH, USA) for 60 min at 37 °C. Then, the samples were alkylated with 55 mM iodoacetamide (IAM, Sigma-Aldrich Co. LLC, St. Louis, MA, USA), protected from light, for 45 min at ambient temperature. The concentration of protein was determined using the Bradford assay, and 5 mg protein per biological replicate was digested. The samples were diluted with 30 mM HEPES (Sigma-Aldrich Co. LLC, St. Louis, MA, USA) until the concentration of urea becomes < 2 M. Trypsin (Promega, Madison, WI, USA) was added into each sample at an enzyme to protein ratio of 1:50 and the samples were further digested overnight at 37 °C. Enzymatic digestion was terminated by adding 0.5% (*v*/*v*) formic acid. Finally, peptides of each sample were desalted and concentrated using Sep-Pack C_18_ Cartridges (Waters, Worcester, MA, USA).

### 2.5. Enrichment of Acetylpeptides

The agarose-conjugated anti-acetyl-lysine antibody (Cell signaling technology, Danvers, MA, USA) was used to enrich the acetylated peptides. Briefly, the antibody beads were thoroughly mixed and resuspended in PBS. After centrifugation (2000× *g*, 30 s, 4 °C), the antibody beads were pre-washed with 1 mL PBS for three times and then with 1 mL of immune affinity purification (IAP) buffer (50 mM MOPS/NaOH, 10 mM Na_2_HPO_4_, 50 mM NaCl, pH 7.2) twice. The treated peptides were dissolved in IAP buffer to a concentration of 1 μg/μL and the standard acetylated peptide (GL Biochem, Minhang District, Shanghai, China) was added into the dissolved peptides as a quality control. Then, the treated peptides were blended with pre-washed antibody beads and incubated with rotary shaking at 4 °C overnight. After incubation, the peptide-antibody mixtures were pelleted at 2000× *g* and 4 °C for 30 s and the supernatant was discarded. The bound peptides were washed twice with IAP buffer and then triced with ice-cold water. Subsequently, the peptides were eluted from the beads two times with 0.15% trifluoroacetic acid (TFA, J&K Scientific, Chaoyang District, Beijing, China) in water. Finally, the eluted peptides were combined and dried under a vacuum according to the manufacturer’s recommendation and used for mass spectrometry analysis.

### 2.6. HPLC and LC-MS/MS

All samples used for MS1 filtering experiment were separated by an HPLC system (Ultimate 3000 UHPLC, Thermo Scientific, Chelmsford, MA, USA) connected to a Q Exactive mass spectrometer (Thermal Scientific, Chelmsford, MA, USA). Peptides were resuspended with phase A (2% acetonitrile (can), 0.1% formic acid (FA)) and centrifuged at 20,000× *g* for 10 min. The supernatant was loaded on the trap column to be enriched and desalted. Then, the peptides were separated at a flow rate of 300 nL/min on a 25 cm analytical column (C18 Acclaim PepMap 100, 75-μm ID × 15 cm, 3-μm particle size, 100-Å pore size; Dionex, Thermal Scientific, Chelmsford, MA, USA) connected to the trap column. The linear gradient of LC was set at 5% buffer B (95% ACN, 0.1% FA) (from 0 to 5 min), 5–25% buffer B (from 5 to 90 min), 25–35% buffer B (from 90 to 100 min), 35–80% buffer B (from 100 min to 110 min), at 80% buffer B (from 110 to 115 min), and a return to 5% buffer B (from 115 min to 116 min), and finally a hold at 5% buffer B (116 to 120 min).

Peptides were ionized by a nano-electrospray ion source and then identified by the Q Exactive Mass Spectrometer (Thermal Scientific, Chelmsford, MA, USA) in the mode of DDA (data-dependent acquisition). The scan of first-grade MS ranged from 350 to 1500 m/z at a resolution of 120,000 and an automatic gain control (AGC) target of 10^6^. The scan of second-grade MS was initiated as 100 m/z at a resolution of 30,000 with a dynamic exclusive time of 30 s and an AGC target of 5 × 10^4^. The mode of second-grade MS spectra was high-energy collisional dissociation (HCD), and the top 10 parent ions with a charge from 2+ to 6+ were selected.

### 2.7. Data Processing and Peptide Identification

Raw mass spectral data were searched against *T. gondii* ME49 strain database (ToxoDB-33_*T. gondii* ME49_Annotated Proteins.fa; 8322 sequences) using MaxQuant software (ver. 1.5.3.30, Max-Planck Institute of Biochemistry, Department of Proteomics and Signal Transduction, Munich, Germany). Parameters of the database search included: Enzyme = trypsin/P, minimal peptide length = 7, PSM-level FDR = 0.01, protein-level FDR = 0.01, fixed modification: carbamidomethyl (C), and variable modification: Oxidation (M), acetyl (protein N-term), acetyl (K). Then, the acetylated sites were filtrated at the level of site decoy fraction ≤ 1% to obtain the significant modification. The *p*-value for identification and quantification of proteins was set as *p* ≤ 0.05 and acetylated proteins with a fold-change of two were deemed as differentially expressed acetylated proteins (DAPs).

### 2.8. Data Analysis and Presentation

Sequence windows of ±5 amino residues adjacent to the acetylation site of each strain were applied to search for conserved motifs using the Motif-X software (ver. 1.2, Harvard Medical School, Boston, MA, USA, http://motif-x.med.harvard.edu/). Each motif was displayed as a logo-like graph. Functional annotation of the differentially acetylated proteins using gene ontology (GO) enrichment analysis (ver. 4.0, Open Biological and Biomedical Ontology Foundry, San Francisco, CA, USA, http://www.geneontology.org) was performed to identify enriched cellular components, biological processes, and molecular functions in each comparison group between two strains. Moreover, Kyoto Encyclopedia of Genes and Genomes (KEGG) analysis (ver. 89.1, Institute for Chemical Research, Kyoto University, Kyoto, Japan, http://www.genome.jp/kegg/) was performed to annotate pathways. GO terms and KEGG pathways with a *p*-value ≤ 0.05 were considered significantly enriched. The protein–protein interaction (PPI) information regarding the acetylated proteins identified in the parasite strains (RH, PRU, and PYS) acetylated proteins identified in the parasite strains (RH, PRU, and PYS) were obtained via searching the online STRING database (Search Tool for the Retrieval of Interacting Genes/Protein, ver. 11.0, Wellcome Genome Campus, Hinxton, Cambridgeshire, UK, http://string-db.org/). The predicted PPI information for the acetylated proteins with a confidence score above 0.7 was selected to construct the PPI networks using the Cytoscape software (ver. 3.2.0, National Institute of General Medical Sciences, National Institutes of Health, Bethesda, MD, USA).

Data and Material Availability: All the mass spectrometry data have been submitted to the ProteomeXchange Consortium with the identifier PXD015031.

## 3. Results

### 3.1. Identification of Lysine Acetylation in RH and PYS Strains

To comprehensively characterize the acetylome of three strains of *T. gondii* (RH, PRU and PYS) purified *in vivo* and to gain new insight into the intergenotype differences in the level of their lysine acetylation, total protein was extracted from purified tachyzoites of RH (type I) strain, PRU (type II) strain, and PYS (Chinese 1) strain *in vivo* and lysine-acetylated peptides were enriched using an immune-affinity approach. Then, the enriched acetylated peptides and acetylation sites were analyzed using a highly sensitive LC-MS/MS technique and bioinformatic analysis.

After manual removal of acetylated peptides with unclear MS/MS spectra, 550 proteins acetylated at 1071 unique sites and 590 proteins acetylated at 1,130 unique sites were detected in replicate one and replicate two of RH strain, respectively. Among these acetylated proteins, 457 acetylated proteins at 961 unique sites were commonly identified in the two replicates (Figure 1A). For PYS strain, replicate one and replicate two included 250 proteins acetylated at 370 unique sites and 239 proteins acetylated at 353 unique sites, respectively. Of these, 188 proteins acetylated at 298 unique sites were shared between the two replicates (Figure 1B). The acetylated proteins simultaneously identified in two replicates of RH strain and PYS strain are listed in Appendix A, respectively.

### 3.2. Sequence Motif Analysis of Acetylation Sites in RH Strain and PYS Strain

In order to characterize the pattern of lysine acetylation sites and identify potential motifs in acetylated proteins in RH and PYS strains, the software motif-x was used to analyze amino acid sequences from the −5 to +5 positions of the identified lysine acetylation sites [31,32]. Among 458 acetylated proteins in RH strain, four acetylation site motifs were enriched from 961 unique acetylation sites, namely xxxxxK^Ac^Hxxxx, xxxxxK^Ac^Fxxxx, xxxxxK^Ac^Nxxxx and xxxxGK^Ac^Sxxxx (K^Ac^ denotes the acetylated lysine, and ‘x’ indicates a random amino acid residue, Figure 2A). Of the 298 unique sites of 188 acetylated proteins in PYS strain, the same number and types of motifs identified in RH strain were also identified around the acetylation sites, including xxxxxK^Ac^Hxxxx, xxxxxK^Ac^Fxxxx, xxxxxK^Ac^Nxxxx and xxxxGK^Ac^xxxxx (Figure 2B). In these motifs, several amino acid residues were conserved, for instance, histidine (H), phenylalanine (F), and asparagine (N) were located downstream of acetylated lysines. While the enrichment of amino acid residues H, F and N were observed in the + 1 position, and amino acid residue glycine (G) was observed in the −1 position.

### 3.3. Gene Ontology (GO) Enrichment of Acetylated Proteins in RH and PYS Strains

To further characterize the function of the acetylated proteins in the two examined strains, GO enrichment analysis was conducted to functionally categorize the acetylome dataset of each strain. The top 10 enriched GO terms with the largest abundant acetylated proteins in the three categories, molecular function (MF), biological process (BP) and cellular component (CC), are shown in Figure 3.

The 458 acetylated proteins in RH strain were annotated into 39 GO terms including 10 ‘CC’ terms, 9 ‘MF’ terms and 20 ‘PB’ terms. The top five enriched ‘CC’ GO terms were cell, cell part, organelle, macromolecular complex, and organelle part. Catalytic activity, binding, structural molecule activity, transporter activity, and electron carrier activity represented the top five most enriched ‘MF’ GO terms. Metabolic process, cellular process, single-organism process, biological regulation and regulation of biological process were the largest enriched five ‘BP’ GO terms.

In terms of PYS strain, the 188 acetylated proteins were enriched into 10 ‘CC’, 8 ‘MF’ and 18 ‘BP’ GO terms, respectively. The five most enriched ‘CC’ GO terms were cell, cell part, organelle, macromolecular complex and organelle part. The top enriched ‘MF’ GO terms, included catalytic activity, binding, structural molecule activity, transporter activity, antioxidant activity, electron carrier activity, molecular transducer activity and receptor activity. The top three most significantly enriched ‘BP’ GO terms included metabolic process, cellular process, and single-organism process.

### 3.4. KEGG Pathway of the Acetylated Proteins in RH and PYS Strains

Acetyl-lysine is known to play a role in the regulation of central metabolic pathways in various organisms. To gain further insights into the metabolism regulation of protein acetylation in the two analyzed *T. gondii* strains, acetylated proteins in RH and PYS strains were mapped to KEGG pathways. The acetylated proteins in RH and PYS strains were mapped to 246 and 170 pathways, respectively. As shown in Figure 4A, the top five pathways with the most significantly enriched acetylated proteins in RH strain were metabolic pathways, biosynthesis of secondary metabolites, biosynthesis of antibiotics, microbial metabolism in diverse environments, and ribosome. As for PYS strain, the five most overrepresented pathways with the highest number of enriched acetylated proteins included: Metabolic pathways, biosynthesis of secondary metabolites, biosynthesis of antibiotics, carbon metabolism, and microbial metabolism in diverse environments (Figure 4B).

### 3.5. Protein–Protein Interaction Networks of the Acetylated Proteins in RH and PYS Strains

The STRING 9.1 database (Search Tool for the Retrieval of Interacting Genes) was searched for protein–protein interaction (PPI) networks of the acetylated proteins in RH and PYS strains. The Cytoscape software was used to construct PPI networks, where nodes represent proteins and edges represent physical interactions. To identify the key molecules, networks for RH and PYS strains were constructed and analyzed in order to identify the major hub proteins (Figure 5 and Figure 6). After removal of duplicated edges and self-loops, the PPI network of RH strain contained 274 protein nodes and 2011 interactor edges, with the major/hub nodes including ribosomal protein RPL5 (TGME49_120050), polyubiquitin UbC (TGME49_019820), ribosomal protein RPS6 (TGME49_010690), ribosomal protein RPS12 (TGME49_005340), and ribosomal protein RPS4 (TGME49_007440) (Figure 5). The PPI network of PYS strain contained 114 protein nodes, which were connected by 479 interactor edges (Figure 6). The hub nodes in this PPI network included polyubiquitin UbC (TGME49_019820), ribosomal protein RPL5 (TGME49_120050), translation elongation factor 2 family protein (TGME49_005470), ribosomal protein RPS12 (TGME49_005340), and ribosomal protein RPL3 (TGME49_027360).

### 3.6. Identification of Lysine Acetylation in PRU Strain

In PRU strain, 134 proteins acetylated at 197 unique sites were characterized in replicate one and 198 proteins acetylated at 297 unique sites were characterized in replicate two. There were 115 proteins acetylated at 190 unique sites commonly found in the two replicates (Figure 7A). The acetylated proteins simultaneously identified in two replicates of PRU strain are listed in Appendix A. Among the overlapped acetylated proteins between two replicates of RH strain, PRU strain, and PYS strain, 83 proteins were simultaneously acetylated at 171 unique sites (Figure 7B). The detailed information regarding overlapped acetylated proteins among the three strains are presented in Appendix A. There were 273, 8, and 16 acetylated proteins exclusively expressed in RH strain, PRU strain, and PYS strain, respectively.

### 3.7. Sequence Motif Analysis of Acetylation Sites in PRU Strain

In order to characterize the pattern of lysine acetylation sites and identify potential motifs in acetylated proteins in PRU strain, the software motif-x was used to analyze amino acid sequences from the −5 to +5 positions of the identified lysine acetylation sites [31,32]. In PRU strain, the three conserved motifs matched with 190 unique sites of 115 acetylated proteins were xxxxGK^Ac^xxxxx xxxxxK^Ac^Fxxxx and xxxxxK^Ac^Hxxxx (Figure 8).

### 3.8. Gene Ontology (GO) Enrichment of Acetylated Proteins in PRU Strain

For GO analysis of PRU strain, the 115 acetylated proteins were associated with 10, 6, and 18 ‘CC’, ‘MF’ and ‘BP’ GO terms (Figure 9). Cell and cell part were the most enriched ‘CC’ GO terms. The most enriched ‘MF’ GO term in molecular function was binding and the metabolic process held the most enriched ‘BP’ GO term.

### 3.9. KEGG Pathway of the Acetylated Proteins in PRU Strain

To gain further insights into the metabolism regulation of protein acetylation in PRU strain, acetylated proteins in PRU strain were mapped to KEGG pathways. The acetylated proteins in PRU strain were mapped to 148 pathways. Metabolic pathways, biosynthesis of antibiotics, biosynthesis of secondary metabolites, carbon metabolism, microbial metabolism in diverse environments were the top five pathways that included the most enriched acetylated proteins (Figure 10). The results of KEGG analysis showed that the metabolic pathways occupied the top position among the identified pathways with the most enriched acetylated proteins in all three strains. Interestingly, acetylated proteins from the virulent strains (RH and PYS) were more enriched in pyruvate metabolism pathway, compared to acetylated proteins from PRU strain.

### 3.10. Protein–Protein Interaction Networks of the Acetylated Proteins in PRU Strain

The STRING 9.1 database (Search Tool for the Retrieval of Interacting Genes) was searched for protein–protein interaction (PPI) networks of the acetylated proteins in PRU strain. The Cytoscape software was used to construct PPI networks, where nodes represent proteins and edges represent physical interactions. To identify the key molecules, networks for PRU strain were constructed and analyzed in order to identify the major hub proteins (Figure 11). After removal of duplicated edges and self-loops, the PPI network of PRU strain contained 67 protein nodes and 193 interactor edges, with the major/hub nodes including polyubiquitin UbC (TGME49_019820), phosphoglycerate kinase PGKI (TGME49_118230), peptidyl-prolyl cis-trans isomerase (TGME49_083850), actin ACT1 (TGME49_009030), and ribosomal protein RPL5 (TGME49_120050).

### 3.11. Quantitation of the Acetylated Proteins Simultaneously Expressed in RH and PYS Strains

A label-free approach was used to quantify the abundance of differentially acetylated proteins (DAPs) simultaneously expressed in RH and PYS strains (|Log2 fold change| >1 and *p* < 0.05). We identified 26 DAPs in PYS strain vs. RH strain (Appendix A). Among the DAPs, 2 acetylated proteins were upregulated, and 24 acetylated proteins were downregulated (e.g., histone acetyltransferase and glycyl-tRNA synthetase) in PYS strain compared to RH strain (Appendix A).

### 3.12. Quantitation of the Acetylated Proteins Simultaneously Expressed in RH, PRU Strains and PRU, PYS Strains

A label-free approach was used to quantify the abundance of differentially acetylated proteins simultaneously expressed in RH, PRU strains and PRU, PYS strains (|Log2 fold change| >1 and *p* < 0.05). We identified 15 DAPs in RH strain vs. PRU strain and 3 DAPs in PRU strain vs. PYS strain (Appendix A). Among the DAPs, there are 10 upregulated acetylated proteins (e.g., histone acetyltransferase and glycyl-tRNA synthetase) and five downregulated acetylated proteins in RH strain vs. PRU strain (Appendix A). Compared with PRU strain, two proteins and one protein were up-acetylated or down-acetylated in RH strain, respectively (Appendix A).

### 3.13. Enrichment Analysis of the DAPs Simultaneously Expressed in RH and PYS Strains

To obtain more insights into the function of the DAPs simultaneously expressed in strains RH and PYS, GO enrichment and KEGG pathway analyses were conducted. Comparing PYS to RH, 40 MF terms, 70 BP terms, and 16 CC terms were identified. The top 10 enriched GO terms with the most abundant DAPs under categories MF, BP and CC in PYS vs. RH are shown in Appendix A. Cell, cell part, intracellular, and intracellular part were the mostly enriched ‘CC’ GO terms. The largest enriched GO terms in BP and MF are cellular process and binding, respectively (Appendix A).

As shown in Appendix A, longevity regulating pathway, multiple species, estrogen signaling pathway, cGMP-PKG signaling pathway, calcium signaling pathway, and aminoacyl-tRNA biosynthesis were the five significantly enriched pathways in PYS vs. RH, and included 2, 2, 3, 3 and 4 DAPs, respectively.

### 3.14. Enrichment Analysis of the DAPs Simultaneously Expressed in RH, PRU Strains and PRU, PYS Strains

To obtain more insights into the function of the DAPs simultaneously expressed in RH, PRU strains and PRU, PYS strains, GO enrichment and KEGG pathway analyses were conducted. We identified 24 MF terms, 42 BP terms and 20 CC terms when comparing RH to PRU, and 9 BP terms and 5 CC terms when comparing PRU to PYS. In the comparison between RH and PRU, the most significantly enriched ‘CC’ GO terms were intracellular part, intracellular, cell and cell part. Metabolic process and catalytic activity occupied the top position of GO terms involved in BP and MF, respectively (Appendix A). With regard to PRU vs. PYS, each of the five ‘CC’ GO terms contained two DAPs. Nine ‘BP’ GO terms are enriched in two DAPs (Appendix A). KEGG analysis identified 2 DAPs that are significantly enriched in HIF-1 signaling pathway in RH vs. PRU (Appendix A).

## 4. Discussion

Little is known about whether intergenotype variation in lysine acetylation contributes to the differences in the virulence of *T. gondii* strains. In an effort to partially address this research need and improve the understanding of the molecular mechanisms underpinning pathogenicity differences between genotypes of *T. gondii,* we studied the acetylomes of three *T. gondii* strains belonging to different genotypes, namely RH (type I) strain, PRU (type II) strain and PYS (Chinese 1) strain. Specifically, we investigated the differences in the level of lysine acetylation within *T. gondii* genotypes using label-free quantitative acetylomics. Our analysis revealed large differences in the acetylation between RH, PRU, and PYS strains. RH strain had the largest level of lysine acetylation, followed by PYS strain, whereas the smallest level of lysine acetylation was detected in PRU strain.

The virulence process of *T. gondii* is dependent on several factors that directly impact several key aspects of the parasite pathogenesis, such as parasite gliding, host cell attachment, invasion, modulation of host response, parasite growth, egress, stage differentiation, cyst formation and parasite burden [33]. In the present study, some virulence factors containing acetylation sites are identified in RH strain, PRU strain and PYS strain (Table 1). Several microneme (MIC) proteins were found to be acetylated in RH strain. Most of the MIC proteins are adhesins or surface proteins responsible for host cell attachment and parasite motility. For example, deletion of MIC1 reduced the ability of *T. gondii* to invade the host cell [34]. MIC2 plays an essential role in gliding and cell invasion [35]. Although MIC3 is dispensable for *in vitro* invasion of *T. gondii*, the virulence of parasites with double MIC1-MIC3 deletion was attenuated *in vivo* [34]. Deletion of MIC4 resulted in mutant parasites with the same phenotype as MIC1 knocked-out [36]. MIC8 not only plays a role in the invasion of *T. gondii* but is also essential in the signaling cascade leading to rhoptry exocytosis [37]. The Apical membrane antigen 1 (AMA1) identified in PYS strain is involved in the formation of moving junction during host cell invasion and intracellular tachyzoite multiplication [38,39]. The preferential frequency of amino acids at positions surrounding acetylated lysine residues was reported in several eukaryotic and prokaryotic organisms [22,30,40,41,42,43,44]. To further explore the differences in the acetylated lysines between the three strains, motif analysis was performed to detect the conserved sequences around the lysine-acetylated sites from acetylation data of RH, PRU and PYS strains. Four putative lysine acetylation motifs (xxxxxK^Ac^Hxxxx, xxxxxK^Ac^Fxxxx, xxxxxK^Ac^N xxxx and xxxxGK^Ac^Sxxxx) and four motifs (xxxxxK^Ac^Hxxxx, xxxxxK^Ac^Fxxxx, xxxxxK^Ac^Nxxxx and xxxxGK^Ac^xxxxx) were identified in the acetylomes of RH strain and PYS strain, respectively. Three motifs (xxxxGK^Ac^xxxxx, xxxxx K^Ac^Fxxxx and xxxxxK^Ac^Hxxxx) were identified in the acetylomes of PRU strain. Three amino acids, including histidine [H] and phenylalanine [F] enriched in the +1 position, and glycine [G]) enriched in the −1 position, were detected in the RH, PRU and PYS strains. However, asparagine (N) was only present in the virulent RH and PYS strains (Figure 2). The motifs are essential for the recognition of substrate by acetyltransferase [30]. Conserved motifs share similar functions among species and different motifs imply different properties. Therefore, the discrepancies of motifs in acetylated proteins in RH, PRU and PYS strains suggest that distinct abilities for substrate recognition by acetyltransferase exist between different *T. gondii* strains. Interestingly, xxxxxK^Ac^Nxxxx seems to play a role in acetylation in virulent strains (RH and PYS) because it was absent in PRU strain. On the other hand, xxxxxK^Ac^Hxxxx, xxxxxK^Ac^Fxxxx, and xxxxGK^Ac^xxxxx were detected in the three *T. gondii* strains (RH, PRU and PYS), suggesting that they might be functionally important for acetylation to occur in *T. gondii*, regardless of the strain’s virulence. These results shed new light on the differences and similarities in substrate recognition by acetyltransferases among RH, PRU and PYS strains.

Preference for histidine (H) at the +1 position of the lysine acetylation site has been observed in acetylated mitochondrial proteins in prokaryotes [41]. Motif xxxxxK^Ac^Hxxxx was detected in the three *T. gondii* strains, suggesting that mitochondrial acetylation in RH, PRU and PYS strains is probably catalyzed by similar acetyltransferases. Another motif, xxxxxK^Ac^Fxxxx, was also identified in RH, PRU and PYS strains. Phenylalanine (F) is the most conserved amino acid surrounding the lysine acetylation site and is an essential precursor of many secondary metabolites [45]. This finding is consistent with the result of functional gene ontology (GO) analysis, where biosynthesis of secondary metabolites in RH, PRU, and PYS strains was found among the top five pathways with the most significantly enriched acetylated proteins. Motif xxxxxK^Ac^Nxxxx (asparagine) was found in RH strain and PYS strain but was absent in PRU strain. *T. gondii* can produce asparagine de novo, however this parasite seems to also acquire asparagine form the host cells. Thus, the presence of motif xxxxxK^Ac^Nxxxx in the virulent strains is anticipated because amino acids are crucial nutrients for *T. gondii* and required for protein synthesis and generation of metabolic intermediates essential for the parasite survival and proliferation. Further support to this observation stems from the finding that metabolic process and catalytic activity occupied the top position of GO terms involved in BP and MF, respectively, when comparing the virulent RH strain to the less virulent PRU strain.

Enzymes with K^Ac^ can regulate central metabolic pathways in various organisms [18,46]. Pyruvate homeostasis plays a key role in the growth and metabolic plasticity of *T. gondii* [47]. The less representation of the pyruvate metabolism pathway in the acetylome of PRU strain may contribute to its being less virulent compared to the virulent strains (RH and PYS). In *T. gondii*, a number of essential functions, such as parasite motility, cell invasion, protein secretion, and egress, are controlled by calcium signaling pathway [48]. When comparing PYS strain to RH strain, the calcium signaling pathway was found enriched in three down-regulated acetylated proteins (TGME49_263300, TGME49_257680, TGME49_230420). The downregulation of the calcium signaling pathway in PYS strain compared to RH strain probably accounts for the slight difference in the virulence between PYS strain and RH strain. Although both RH and PYS are virulent strains, PYS strain seems to exhibit slightly less virulence than that of RH strain

We identified 26 DAPs in PYS strain vs. RH strain. 15 and 3 DAPs were identified in RH strain vs. PRU strain and PRU strain vs. PYS strain, respectively. Intriguingly, histone-acetyl-transferase and glycyl-tRNA synthase had higher expression levels in RH strain than in PYS strain (Appendix A) and PRU strain (Appendix A). Histone acetylation is associated with stage-specific gene expression in *T. gondii*. RH strain deficient in histone-acetyl-transferase had reduced sensitivity toward high-pH and reduction in the expression of about 74% of stress-response-related genes compare to wild-type strains [49]. Moreover, strains lacking histone-acetyl-transferase did not only exhibit a reduction in the expression of key genes regulating cyst-form, but also their ability to remain active under high-pH conditions was diminished. High-pH was usually used for the induction of *T. gondii* stage phenotypic transformation *in vitro*, thus histone-acetyl-transferase plays a key role in *T. gondii* response to high-pH [49]. Therefore, the high expression of histone-acetyl-transferase in RH strain reflects the difference between *T. gondii* genotypes in terms of tolerance to external stress. Likewise, increased expression of glycyl-tRNA synthetase in RH strain compared to PRU strain may contribute to the increased ability of RH strain to proliferate, a key feature in the parasite’s virulence. Glycyl-tRNA synthetase plays a crucial role in protein synthesis and in the neddylation of proteins in eukaryotic cells [50]. Neddylation is a type of protein PTM mediated by conjugation of the ubiquitin-like protein Nedd8 to specific substrates and regulates protein function and cell cycle progression. The protein neddylation process has been reported in *Trypanosoma brucei* [51,52,53,54]. Whether neddylation also occurs in *T. gondii* and what exact role does glycyl-tRNA synthetase play in the PTM of *T. gondii* remains to be investigated. Taken together, we used label-free quantitative acetylomics approach to identify differences in the acetylomes in *T. gondii* strains of different genotypes. We detected a large number of acetylated proteins in the tachyzoites of RH (type I) strain, PRU (type II) strain and PYS (Chinese 1) strain. The DAPs between different strains were enriched in various biological functions and pathways. These data expand our knowledge of lysine acetylation in *T. gondii* and may assist in further understanding of the mechanisms that underpin different virulence abilities of *T. gondii*. Further molecular biological experiments and computational investigations are warranted to elucidate the role of the identified acetylated proteins in mediating in the interaction between *T. gondii* and host cell.

## Figures and Tables

**Figure 1 microorganisms-07-00510-f001:**
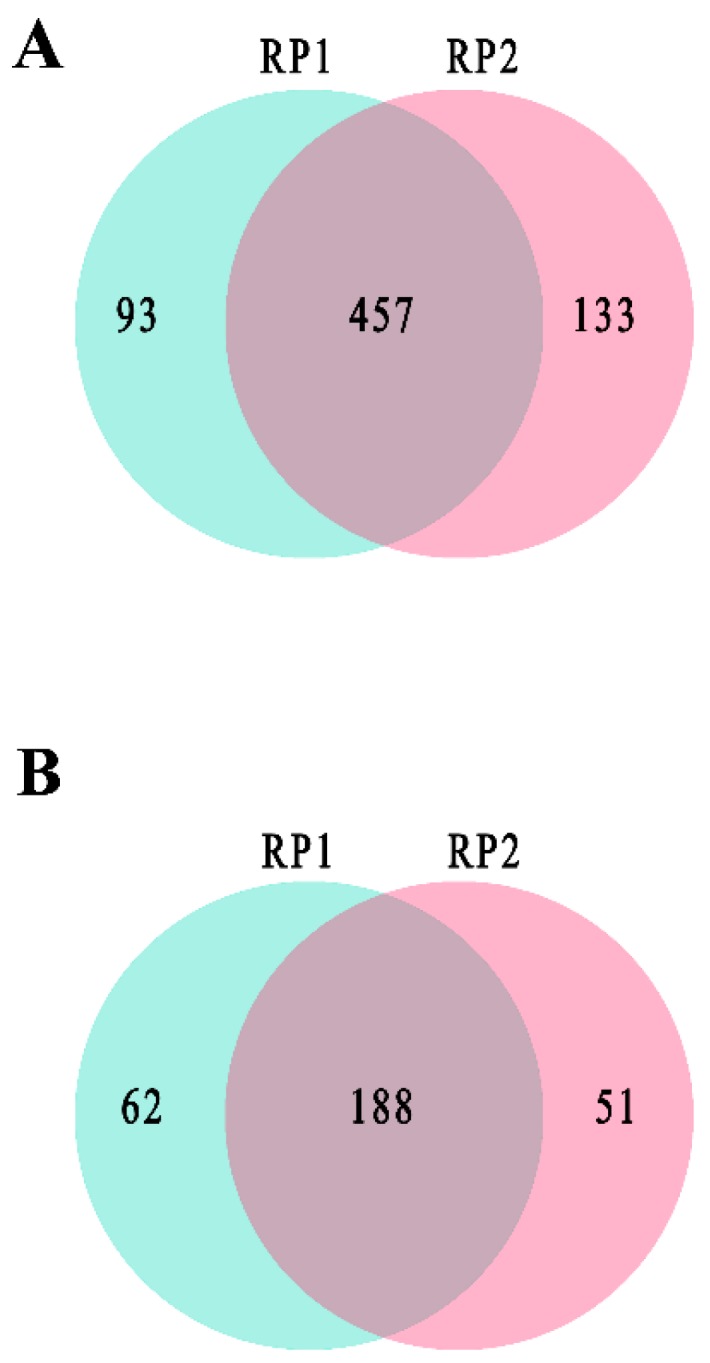
Global identification of lysine acetylation in two virulent strains *T. gondii* RH strain and PYS strain. Venn diagrams show the number of unique and common acetylated proteins in RP1 and RP2 of (**A**) RH strain, (**B**) PYS strain.

**Figure 2 microorganisms-07-00510-f002:**
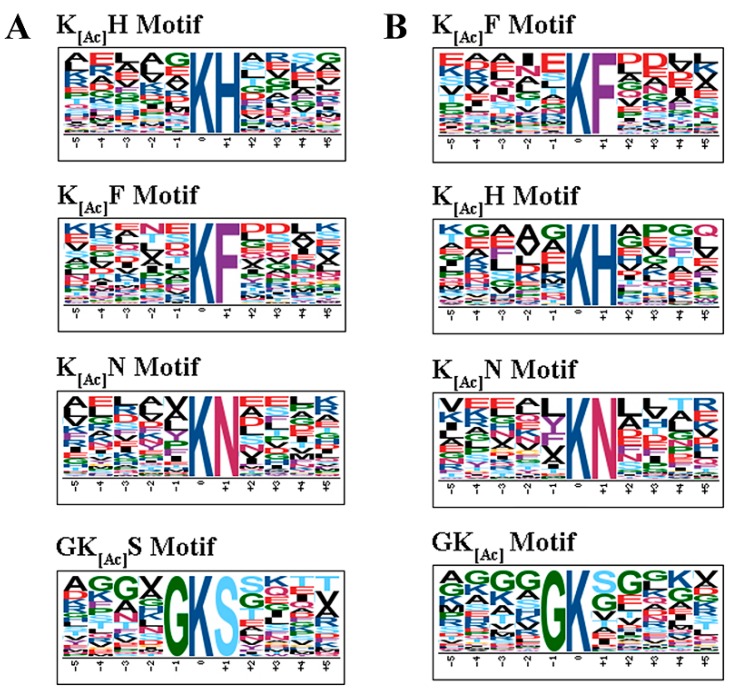
Motif analysis of all identified K_Ac_ acetylated sites in RH strain and PYS strain. Sequence logo representation of significantly enriched acetylation site motifs for ±5 amino acids around the lysine acetylation sites. (**A**) four motifs in RH strain, (**B**) four motifs in PYS strain.

**Figure 3 microorganisms-07-00510-f003:**
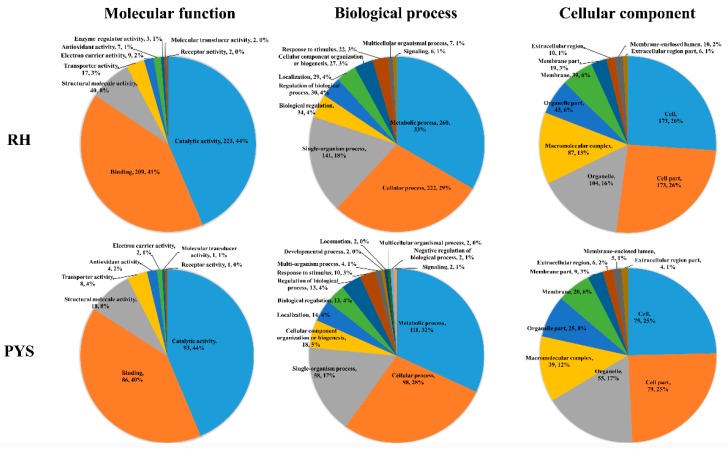
Gene Ontology (GO) enrichment analysis of the acetylated proteins in RH and PYS strains. Functional classification of acetylated proteins in each strain according to the categories of molecular function, biological process, and cellular component. The numbers of DAPs and percentages of DAPs in relation to the total acetylated proteins involved in specific GO terms are shown.

**Figure 4 microorganisms-07-00510-f004:**
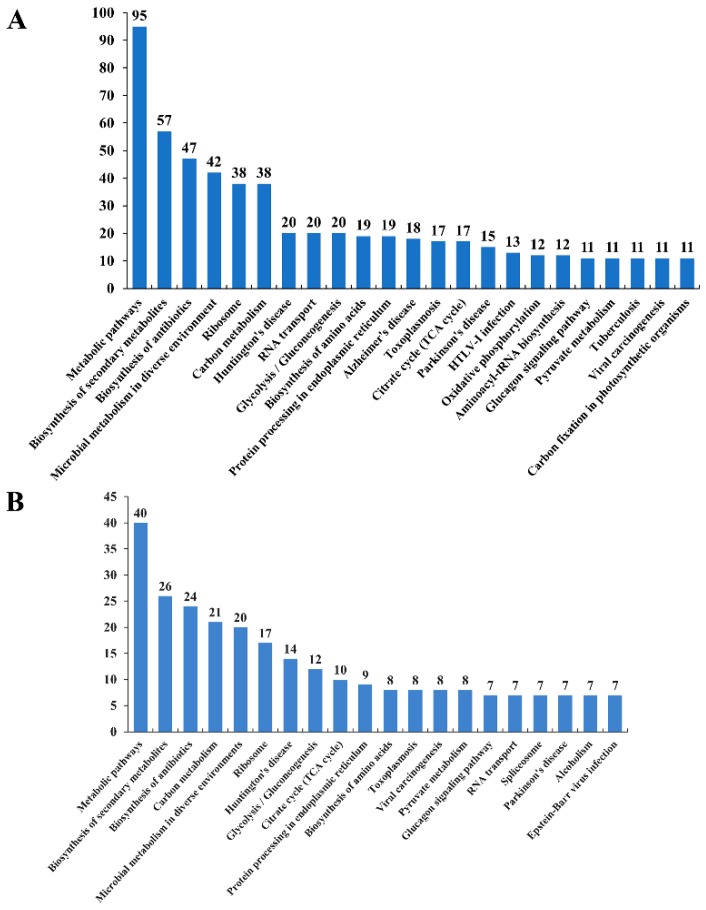
KEGG pathway analysis of the differentially acetylated proteins in RH and PYS strains. Charts (**A**,**B**) represent KEGG pathways of acetylated proteins in RH strain and PYS strain, respectively. The x-axis represents enriched KEGG pathways and the numbers above the bars (y-axis) are the number of acetylated proteins.

**Figure 5 microorganisms-07-00510-f005:**
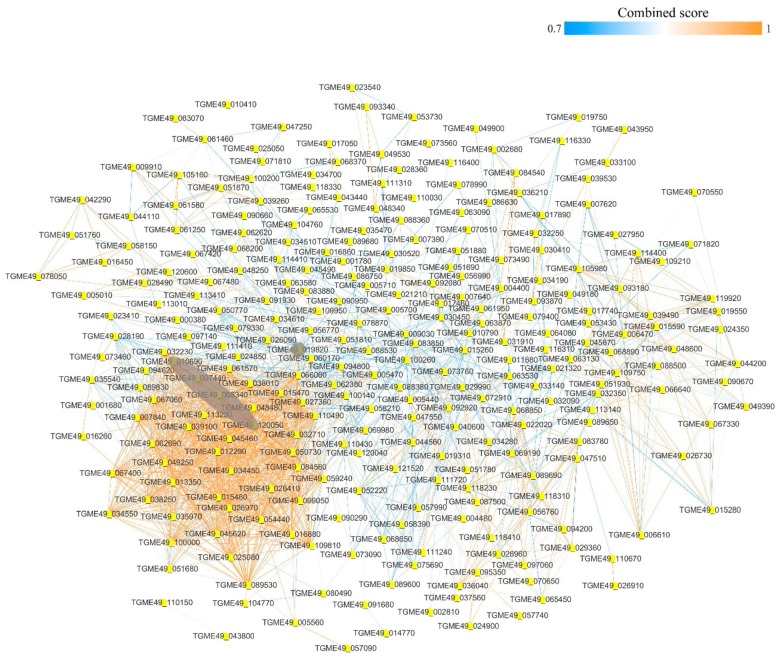
Protein–protein interaction network of the acetylated proteins in RH strain. Nodes represent the acetylated proteins and edges represent interactors between acetylated proteins. The color of the edges denotes the combined score of interactors. The grey color indicates the major/hub nodes of the protein-protein interaction network.

**Figure 6 microorganisms-07-00510-f006:**
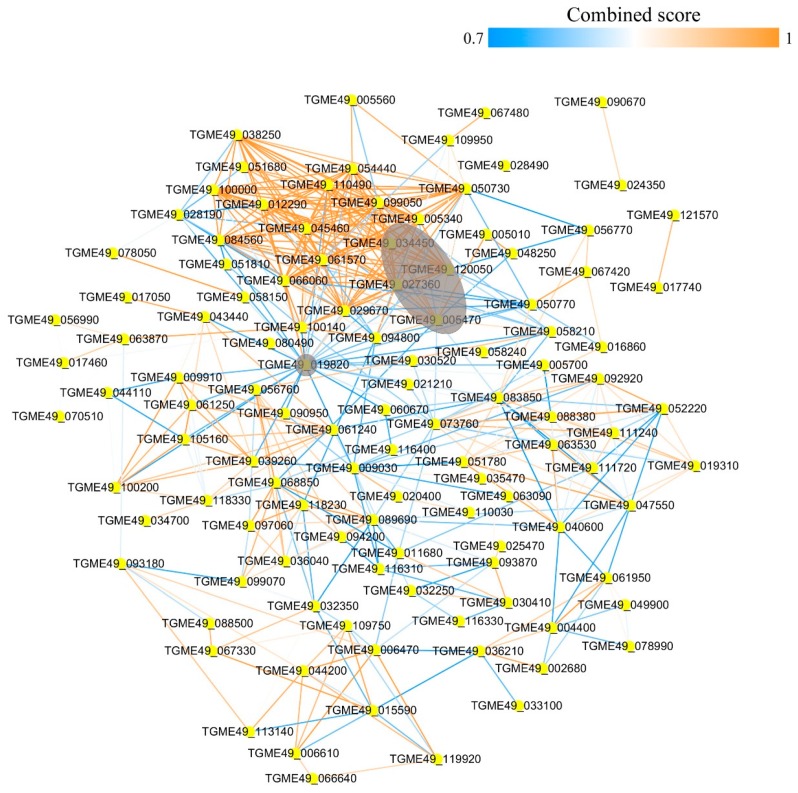
Protein–protein interaction network of the acetylated proteins in PYS strain. Nodes represent the acetylated proteins and edges represent interactors between acetylated proteins. The color of the edge denotes the combined score of interactors. The grey color indicates the major/hub nodes of the protein-protein interaction network.

**Figure 7 microorganisms-07-00510-f007:**
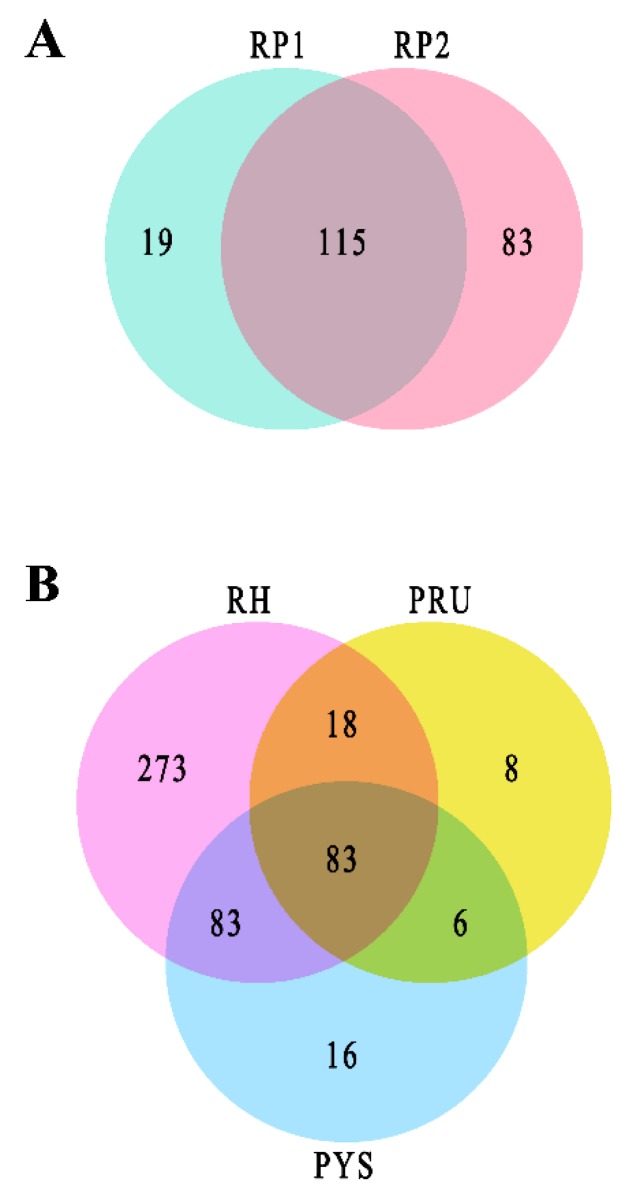
Global identification of lysine acetylation in *T. gondii* PRU strain. Venn diagrams show the number of unique and common acetylated proteins in RP1 and RP2 of (**A**) PRU strain. (**B**) Venn diagram shows the number of unique and common acetylated proteins between RH strain, PRU strain, and PYS strain. RP1 and RP2 indicate biological replicate 1 and biological replicate 2, respectively.

**Figure 8 microorganisms-07-00510-f008:**
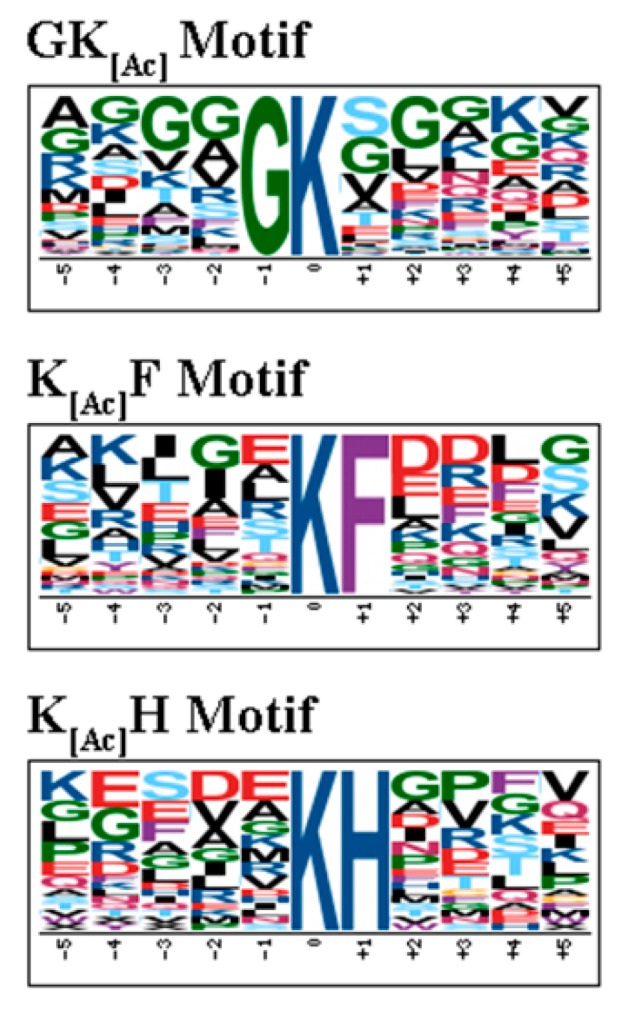
Motif analysis of all identified K_Ac_ acetylated sites in PRU strain. Sequence logo representation of significantly enriched acetylation site motifs for ±5 amino acids around the lysine acetylation sites.

**Figure 9 microorganisms-07-00510-f009:**
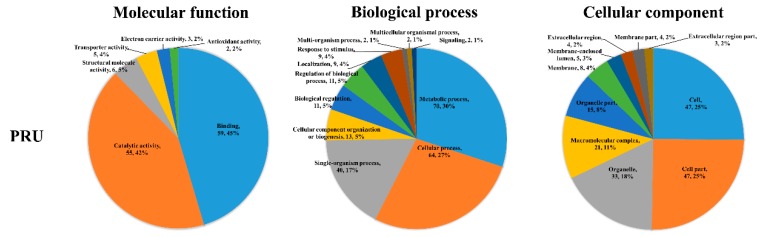
GO enrichment analysis of the differentially acetylated proteins (DAPs) in PRU strain. Functional classification of acetylated proteins in each strain according to the categories of molecular function, biological process, and cellular component. The numbers of DAPs and percentages of DAPs in relation to the total acetylated proteins involved in specific GO terms are shown.

**Figure 10 microorganisms-07-00510-f010:**
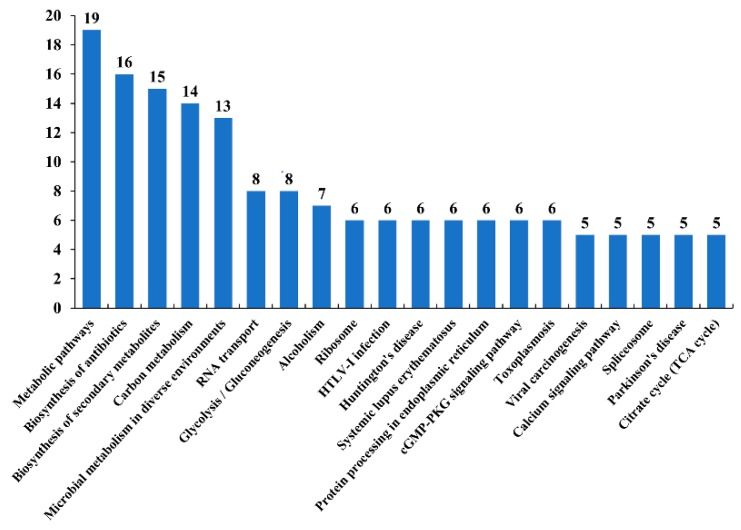
KEGG pathway analysis of the differentially acetylated proteins in PRU strain. The x-axis represents enriched KEGG pathways and the numbers above the bars (y-axis) are the number of acetylated proteins.

**Figure 11 microorganisms-07-00510-f011:**
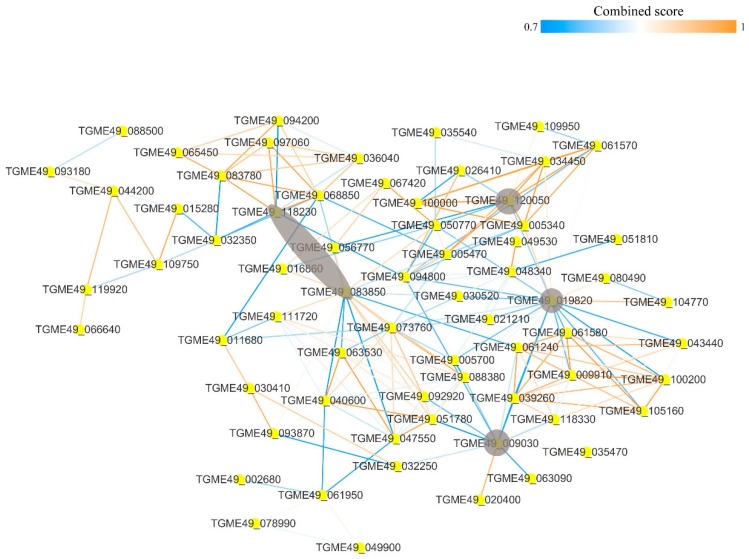
Protein–protein interaction network of the acetylated proteins in PRU strain. Nodes represent the acetylated proteins and edges represent interactors between acetylated proteins. The color of the edges denotes the combined score of interactors. The grey color indicates the major/hub nodes of the protein-protein interaction network.

**Table 1 microorganisms-07-00510-t001:** Virulence factors exclusively acetylated in RH strain, PRU strain and PYS strain.

Protein_ID	Description	Strain
TGME49_231850	Serine-threonine phosophatase 2C (PP2C)	RH
TGME49_291890	Microneme protein (MIC1)	RH
TGME49_227280	Dense granule protein (GRA3)	RH
TGME49_201780	Microneme protein (MIC2)	RH
TGME49_257680	Myosin light chain (MLC1)	RH
TGME49_229010	Rhoptry neck protein (RON4)	RH
TGME49_211290	Rhoptry protein (ROP15)	RH
TGME49_319560	Microneme protein (MIC3)	RH
TGME49_208030	Microneme protein (MIC4)	RH
TGME49_245490	Microneme protein (MIC8)	RH
TGME49_291960	Rhoptry kinase family protein (ROP40)	RH
TGME49_200250	Microneme protein (MIC17A)	RH
TGME49_260820	IMC sub-compartment protein 1 (ISP1)	PRU
TGME49_239740	Dense granule protein (GRA14)	PRU
TGME49_315490	Rhoptry protein (ROP10)	PYS
TGME49_211260	Rhoptry kinase family protein (ROP26)	PYS
TGME49_255260	Apical membrane antigen (AMA1)	PYS
TGME49_280570	SAG-related sequence (SRS35A)	PYS

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
