# Peer review of "Label-Free Quantitative Acetylome Analysis Reveals Toxoplasma gondii Genotype-Specific Acetylomic Signatures"

_microorganisms, 2019, doi:10.3390/microorganisms7110510_

Round 1

Reviewer 1 Report

In section 4.5 last sentence “Finally, the eluted peptides were combined and dried under a vaccum (Sep-Pack C18
39 Cartridges, Waters)() according to the manufacture’s recommendation, and used for mass
40 spectrometry analysis.” Authors need to remove “(Sep-Pack C18 39 Cartridges, Waters)() according to the manufacture’s recommendation,” or indicate if the C18 cartridges are used for desalting and concentration. “manufacture” should be “manufacturer”, and vacuum is spelled incorrectly.

”conflict interests” should be “conflicts of interests”.

Author Response

October 16, 2019

Ms. Norah Tang

Assistant Editor

Microorganisms

Dear Ms. Norah Tang,

Re: Revised Manuscript ID 615490.R2

On behalf of all co-authors, I would like to thank you and the two reviewers very much for further positive comments and constructive suggestions on our manuscript (MS) ID 615490R1.

 We have revised our MS further, strictly according to the reviewers’ comments and suggestions. We used the “tracked changes” mode in the WORD to show the revised/changed text in the revised MS. Two MS files are uploaded: the “clean version” as “manuscript”, and the one showing “tracked changes” as “supplementary material”. In the following, we detail our point-by-point responses to the reviewer’s comments and suggestions. We made all our responses in blue colour for clarity.

Responses to comments and suggestions of Reviewer #1

Comments and Suggestions for Authors

In section 4.5 last sentence “Finally, the eluted peptides were combined and dried under a vaccum (Sep-Pack C18

39 Cartridges, Waters)() according to the manufacture’s recommendation, and used for mass

40 spectrometry analysis.” Authors need to remove “(Sep-Pack C18 39 Cartridges, Waters)() according to the manufacture’s recommendation,” or indicate if the C18 cartridges are used for desalting and concentration. “manufacture” should be “manufacturer”, and vacuum is spelled incorrectly.

”conflict interests” should be “conflicts of interests”.

Response: We thank reviewer #1 very much for constructive suggestions on our manuscript. We have deleted that sentence and corrected the spelling errors accordingly.

We have done our best to address all reviewers’ comments and we sincerely hope that you find our MS revised to your satisfaction. We are looking forward to receiving your editorial decision soon.

With best wishes,

Xing

Xing-Quan Zhu, BVSc, MVSc, PhD

Professor and Head, Department of Parasitology,

Deputy Director, State Key Laboratory of Veterinary Etiological Biology,

Lanzhou Veterinary Research Institute, Chinese Academy of Agricultural Sciences

1 Xujiaping, Yanchangbu, Lanzhou, Gansu Province 730046,

The People's Republic of China

Email: zhuxingquan@caas.cn; xingquanzhu1@hotmail.com

Tel: +86-18793138037; Fax: +86-931-8340977

Reviewer 2 Report

The authors have made a considerable effort in clarifying the present manuscript. However, as stated in my previous review, a comparison of the three strains requires a preparation of the tachyzoites under identical conditions. For strain PRU, this was not the case (dexamethsone treatment of infected mice). Therefore, a direct comparison can be performed only between strains RH and PYS. The presentation of the results should be restructured accordingly. It is unclear why the tachyzoites of all three strains have not been prepared using appropriate and defined tissue culture systems thus preventing potential bias issuing from the isolation from mice.

Author Response

October 16, 2019

Ms. Norah Tang

Assistant Editor

Microorganisms

Dear Ms. Norah Tang,

Re: Revised Manuscript ID 615490.R2

On behalf of all co-authors, I would like to thank you and the two reviewers very much for further positive comments and constructive suggestions on our manuscript (MS) ID 615490R1.

 We have revised our MS further, strictly according to the reviewers’ comments and suggestions. We used the “tracked changes” mode in the WORD to show the revised/changed text in the revised MS. Two MS files are uploaded: the “clean version” as “manuscript”, and the one showing “tracked changes” as “supplementary material”. In the following, we detail our point-by-point responses to the reviewer’s comments and suggestions. We made all our responses in blue colour for clarity.

Responses to comments and suggestions of Reviewer #2

Comments and Suggestions for Authors

The authors have made a considerable effort in clarifying the present manuscript. However, as stated in my previous review, a comparison of the three strains requires a preparation of the tachyzoites under identical conditions. For strain PRU, this was not the case (dexamethsone treatment of infected mice). Therefore, a direct comparison can be performed only between strains RH and PYS. The presentation of the results should be restructured accordingly. It is unclear why the tachyzoites of all three strains have not been prepared using appropriate and defined tissue culture systems thus preventing potential bias issuing from the isolation from mice.

Response: We thank the reviewer very much for favorable comments and constructive suggestions on our revised manuscript.

The strain PRU is an avirulent strain and cyst-forming in mice. It does not form tachyzoites. Therefore, the strain PRU is passaged as cysts in mice in laboratory. In order to harvest and purify tachyzoites of strain PRU, mice were treated with dexamethasone (0.2 mg per mouse) on alternate day and were orally inoculated with 100-150 cysts of PRU strain. This method of isolating tachyzoites of strain PRU in vivo has been widely used (Xu, et al, 2013; Zhou, et al, 2014). Although some researches prepared tachyzoites of T. gondii using tissue culture systems, research on tachyzoites prepared using mice inoculation is interesting and necessary. Because mouse inoculation was the best method for eventual diagnosis of toxoplasmosis and the intraperitoneal is the most generally useful route (Abbas, et al, 1967). The sensitivity of tissue culture and mouse inoculation methods for the purification of tachyzoites of T. gondii were found to be equal (Derouin, et al, 1987). In addition, many researches prepared tachyzoites using mice inoculation (Wang, et al, 2012; Zhang, et al, 2014; Xu, et al, 2013; Zhou, et al, 2014; Wang, et al, 2017). The purpose of our research was to investigate acetylomic signatures of different genotypes of Toxoplasma gondii purified in vivo. So, the tachyzoites of all three strains were isolated from mice, but were not prepared using tissue culture systems. We have restructured results in the revised manuscript strictly according to your comments and suggestions.

We have done our best to address all reviewers’ comments and we sincerely hope that you find our MS revised to your satisfaction. We are looking forward to receiving your editorial decision soon.

With best wishes,

Xing

Xing-Quan Zhu, BVSc, MVSc, PhD

Professor and Head, Department of Parasitology,

Deputy Director, State Key Laboratory of Veterinary Etiological Biology,

Lanzhou Veterinary Research Institute, Chinese Academy of Agricultural Sciences

1 Xujiaping, Yanchangbu, Lanzhou, Gansu Province 730046,

The People's Republic of China

Email: zhuxingquan@caas.cn; xingquanzhu1@hotmail.com

Tel: +86-18793138037; Fax: +86-931-8340977

Round 2

Reviewer 2 Report

The manuscript is now better understandable. The methodology is well presented and clearly merits publication. I still have problems with the direct comparison of strains prepared under different circumstances (i.e. presence and absence of dexamethasone). Perhaps, this point was not clear in my previous reviews. To improve the scientific soundness of the manuscript, I recommend the authors to perform a direct comparison between the two strains prepared without this compound (i.e. RH and PYS) and to present the results with the other strain (i.e. PRU) separately. Consequently, Fig1, the corresponding text and the discussion should be modified accordingly. 

Author Response

October 19, 2019

Ms. Norah Tang

Assistant Editor

Microorganisms

Dear Ms. Norah Tang,

Re: Revised Manuscript ID 615490.R3

On behalf of all co-authors, I would like to thank you and the reviewer #2 very much for further positive comments and constructive suggestions on our manuscript (MS) ID 615490R2.

 We have revised our MS further, strictly according to the reviewers’ comments and suggestions. We used the “tracked changes” mode in the WORD to show the revised/changed text in the revised MS. Two MS files are uploaded: the “clean version” as “manuscript”, and the one showing “tracked changes” as “supplementary material”. In the following, we detail our point-by-point responses to the reviewer’s comments and suggestions. We made all our responses in blue colour for clarity.

Responses to comments and suggestions of Reviewer #2

Comments and Suggestions for Authors

The manuscript is now better understandable. The methodology is well presented and clearly merits publication. I still have problems with the direct comparison of strains prepared under different circumstances (i.e. presence and absence of dexamethasone). Perhaps, this point was not clear in my previous reviews. To improve the scientific soundness of the manuscript, I recommend the authors to perform a direct comparison between the two strains prepared without this compound (i.e. RH and PYS) and to present the results with the other strain (i.e. PRU) separately. Consequently, Fig.1, the corresponding text and the discussion should be modified accordingly.

Response: We thank the reviewer very much for further favorable comments and constructive suggestions on our revised manuscript. We have revised our manuscript strictly according to the reviewer’s comments and suggestions. We have modified Figure 1, the corresponding text and the discussion accordingly.

We have done our best to address all reviewers’ comments and we sincerely hope that you find our MS revised to your satisfaction. We are looking forward to receiving your editorial decision soon.

With best wishes,

Xing

Xing-Quan Zhu, BVSc, MVSc, PhD

Professor and Head, Department of Parasitology,

Deputy Director, State Key Laboratory of Veterinary Etiological Biology,

Lanzhou Veterinary Research Institute, Chinese Academy of Agricultural Sciences

1 Xujiaping, Yanchangbu, Lanzhou, Gansu Province 730046,

The People's Republic of China

Email: zhuxingquan@caas.cn; xingquanzhu1@hotmail.com

Tel: +86-18793138037; Fax: +86-931-8340977